# Aging and the Future of Decent Work

**DOI:** 10.3390/ijerph18178898

**Published:** 2021-08-24

**Authors:** Frida Marina Fischer, Maria Carmen Martinez, Camila Helaehil Alfredo, João Silvestre Silva-Junior, Jodi Oakman, Teresa Cotrim, Donald Fisher, Stephen Popkin, Gretchen A. Petery, Paul A. Schulte

**Affiliations:** 1Department of Environmental Health, School of Public Health, University of São Paulo, São Paulo 01246-904, Brazil; fmfische@usp.br (F.M.F.); camilahelaehil@usp.br (C.H.A.); 2WAF Informatics & Health Ltd., São Paulo 04109-100, Brazil; mcmarti@uol.com.br; 3Department of Medicine, São Camilo University Center, São Paulo 04263-200, Brazil; joao.junior@prof.saocamilo-sp.br; 4Centre for Ergonomics and Human Factors, LaTrobe University, Melbourne, VIC 3086, Australia; j.oakman@latrobe.edu.au; 5Ergonomics Laboratory, Faculdade de Motricidade Humana, University of Lisbon, 1499-002 Lisbon, Portugal; tcotrim@fmh.ulisboa.pt; 6Volpe National Transportation Systems Center, U.S. Department of Transportation, Cambridge, MA 02142, USA; Donald.Fisher@dot.gov (D.F.); Stephen.Popkin@dot.gov (S.P.); 7National Institute for Occupational Safety and Health, Cincinnati, OH 45226, USA; qnd7@cdc.gov

**Keywords:** aging, productive aging, decent work, work ability, work organization

## Abstract

The United Nations identified decent work and economic growth as a sustainable development goal for 2030. Decent work is a term that sums up aspirations for people in their working lives. One of the factors that influences the achievement of decent work is aging. This article examines how aspects of aging and organizational factors affect work ability across the lifespan and throughout one’s work career. Additionally, the critical issue of worker physical mobility was also addressed as a practical limitation to functional aging. Through our investigation, we identified gaps in the literature where research and interventions should be promoted. These include early disability studies; population dashboards of workers’ health metrics; intervention and cost effectiveness in health promotion and prevention of early functional aging at work; policies for tailoring demands to individual needs and abilities; and inequities of social protection for aging workers.

## 1. Introduction

Widespread population aging across the Organisation for Economic Cooperation and Development (OECD) and in many developing and emerging economies represents great achievements in healthcare, medicine, and economic growth, while also introducing new complications [1]. The combination of longer lives, declining birth rates, urbanization, technological changes, and more women entering the workforce have resulted in a rapid change in the age profile of most countries, creating economic challenges [2,3]. Moreover, the impacts of population aging are not equally distributed across countries. For instance, in 2018, Italy, Germany, and Japan reported a median age of 46, compared to the median age in emerging economies such as India, Indonesia, and South Africa, where half the population is under 30. Korea is expected to experience the sharpest rise in median age, from 36 years in 2018 to 43 years in 2050 [1].

To manage such significant shifts in population age distribution requires a range of strategies, including taking steps to encourage and support individuals’ ability to extend their worklife and delay retirement later than previous generations. Determinants of retirement are complex [4] and include a combination of individual (e.g., health) and macro level factors such as policy and the status of the labour market [5]. A comprehensive systems-based approach will be essential to ensure that the push and pull factors towards retirement are addressed in a way that ensure individuals want to continue working and are physically and cognitively able to remain at work [6,7,8].

Aging is an inherent factor that affects the achievement of decent work. The concept of decent work was introduced by the International Labour Organization [9] as a conceptualization of the quality of employment. This multifactor concept is evidenced by equity, dignity, security, and freedom within working conditions [10]. The United Nations has recognized achieving decent work as a suitable development goal for 2030, stating:

“Decent work involves opportunities for work that is productive and delivers fair income, security in the workplace and social protection for families, better prospects for personal development and social integration, freedom for people to express their concerns, ability to organize and participate in decisions that affect their lives, and equality of opportunity for all women and men”[11]

In this article, a diverse group of experienced investigators with expertise in aging, work ability, and decent work examine the relationship between aging, organizational factors, and work ability. This is a commentary based on selected literature identified by the authors as pertinent to the specific topics. In one illustration of difficult situations for workers, we drill down and examine the role of aging in the rapidly changing transportation landscape. With these aspects as a foundation we identify gaps in the published literature about aging and work. Finally, we examine the challenge of achieving decent work as working life increases.

## 2. Work Ability and Individual Work Continuance or Withdrawal

Work ability, a concept based on the perception of the balance between the individual and work resources and demands, evolves across the lifespan and throughout one’s work career [12,13]; it is a meaningful indicator of career sustainability [14]. Individual factors, such as poor health, the presence of disease, work-related musculoskeletal disorders, and stress, are associated with low work ability [15,16]. Importantly, poor health is predictive of withdrawal from the workforce [6]. Workers with low work ability have been identified as more likely to exit the workforce at a younger age than those with high work ability [17]. Furthermore, low work ability can often be identified from mid-life [18]. Aging is commonly associated with a decline in work ability [15,19], but individual changes may differ depending on occupational trajectories, mental and physical strain, and individual resources [14,16].

The impairment of work ability has negative and interconnected effects for individuals, institutions, and society [20]. Poor work ability is associated with increased absence from work, lower job satisfaction, loss of individual and organizational productivity, performance decrement, decreases in career opportunities, higher unemployment, and earlier retirement and exit from careers and the labor force [3,21,22,23].

Functional capacity is a precondition for people to be able to perform personal and job-related tasks and to maintain an active working life. It is also closely related to health status, which is the basis for sustaining work ability [21]. Factors that compromise health and functional capacity can limit prolonging working life [3,21,24]. Approaches by a number of countries have included raising or intentions to raise retirement ages to encourage individuals to remain at work. Another tactic some governments have introduced is introducing gradual-retirement schemes, where workers gradually decrease their work hours, thereby extending working lives while reducing workload [25,26,27]. Policy initiatives such as these are important, but in isolation they are not effective retention strategies without taking into consideration other individual factors [2]. Moreover, analysis of the impact of raised retirement ages remains limited [28], and gradual-retirement options do not have broad appeal to workers [29].

### 2.1. Organizational Factors and Work Ability

In general, work is considered good for health [30], and employment is beneficial to individuals, particularly for reducing depression and improving general mental health [31]. However, this is not universally true. The quality (better/worse) of working conditions contributes to (good/poor) individual health. In this way, work quality is instrumental in sustained employment across the course of individuals’ working lives [32].

There are several work-related demands that influence the decline of work ability. Among them, a highly physical job (e.g., firefighter, construction worker) is a main determinant of increased risk for moderate or poor work ability over time [15,33]. In addition, organizational factors such as low autonomy, job control, social support at work, and reward relative to effort, as well as high work–family conflict, are determinants of worse work ability [34,35,36].

Functional aging at work reflects the reduction of work ability as a result of exposures to workloads throughout the working life and can manifest relatively early, often preceding chronological aging [37,38]. Early functional aging goes beyond the impairment of the functional abilities related to age, because it concerns workers’ difficulties and hindrances in developing their work activities when the physical and mental demands of the work exceed their personal resources [39].

There is tremendous individual variability in functional aging among people of same and different age groups, as well as the rate of change in functional age throughout the lifespan. This creates challenges for developing broad strategies for managing work-related factors that contribute to functional aging. However, two primary considerations are slashing or preventing of years lived with disability (YLD) and years of lost life (YLL) [40]. Preventive actions targeting workers with chronic noncommunicable diseases should be included with occupational safety and health programs. As a result, actions directed at promoting and maintaining work ability are needed for workers of all age groups [21,22,24].

Workplace health promotion programs may encounter obstacles that impede desired results. For example, employers are generally not obliged to promote employee health in the same way they are required to address workplace safety. Lack of resources, management resistance, and employee reluctance to change behaviors are common barriers to program success [41]. The literature on health promotion interventions targeting older workers is sparse but suggests the effectiveness of such programs may be limited and may vary depending on the focus of the intervention [32,42,43,44].

Nevertheless, there are organizational factors that support the maintenance of work ability over time. For instance, offering the ability to adjust working time and take training to improve individual skills and knowledge, having a good sense of community and support at work, and providing meaningful work are positive predictors of a good or excellent work ability [14,45,46]. In terms of workforce aging, one of the major challenges for the future of work concerns the unpredictability of changes in the nature of work that will likely make some individual competencies antiquated in the long run [14], which in turn may increase job insecurity and influence the maintenance of good or excellent work ability. A high level of organizational support has been shown in the past to be positively associated with excellent work ability [34]. As we head into the future, organizational support will likely continue to be an important predictor of work ability.

In managing workers’ occupational health, careful attention should be paid to job design [47]. Emphasis should be placed on optimizing the work environment to enable safe and productive work, while also providing a good fit between workers and their environment [7]. Workloads (physical, cognitive, and adequate work organization) should be adapted to correspond with the individual ability of each worker and should be reviewed and revised at regular intervals. In addition, leaders should develop systemic guidelines for promoting work ability and professional rehabilitation [3,21,22,24]. Encouraging longer working lives requires addressing the gap between good and bad jobs to ensure we have a population with sufficient good health to make choices about the timing of their retirement [48]. To be most effective, these strategies must be underpinned by social protection policies.

Useful management models are available to assist in the systematization of intervention programs aimed at promoting health, work ability, employability, and job retention as people age. Examples include the House Model [49], the action plans of the European Network for Workplace Health Promotion [24], the WHO Healthy Workplace Framework and Model [50], and the Total Worker Health^®^ framework of NIOSH [51]. These models, variably, are based on situational diagnostics and offer underlying best practices including direct priorities and actions; integrated, systematized, and continuous interventions; periodic reevaluations; multi-professional teams; and worker participation. Moreover, they provide guidance for improving working conditions and the organization of work, skills training and development, career plans, improving corporate culture and communication, facilitation of rehabilitation, reintegration after illness or accident, and specific interventions for aging workers. Example interventions include improving design and positioning of the equipment, providing adequate lighting, training on proper lifting procedures and on carrying loads, allowing for longer recovery time between work shifts, improving the work schedules, and allowing rest breaks during shifts [21,24]. To illustrate, a systematic review found interventions following the Total Work Health^®^ approach of combining health protection with health promotion were effective in improving workers’ health and in reducing workplace safety risks [32].

### 2.2. Aging and the Changing Transportation Landscape

The ability to travel to work is critical for functional aging. If a worker’s physical mobility in terms of transportation is limited, the ability to be successful at work may be impaired. Thus, safe transportation is a critical need for successful aging. The risk of being in a fatal crash starts increasing at age 70 and becomes dramatically higher at age 80, on par with the risk for novice drivers [52]. Experience does not always counteract fundamental impairments, making driving increasingly unsafe, if not impossible, for older adults [53]. Functional impairments include not only sensorial and neurological decline but also declines in vision, hearing, and cognition, as are associated with inadequate sleep and increases in medications, both prescribed and over–the–counter. From an occupational health perspective, these issues impact the safety and mobility not only of older adults who must drive to work but also of older adults whose work is driving.

The transportation landscape is rapidly changing. Advances in automation mean more cars, trucks, buses, and other vehicles feature automated driving features. Advances run a spectrum from low to high levels of automation. At the low end there are active safety systems, such as automatic emergency braking and electronic stability control, and driver support features, such as adaptive cruise control (ACC) and lane centering assist (LCA). High level automation includes automated driving features that require no input from the driver [54]. These advances are projected to have three effects on the occupational health of older adults.

First, advances in automation will greatly extend the ways in which older adults can commute to their job, providing transportation to some who may have never had such access, as well as transportation to those whose functional impairments make driving no longer possible [55]. For example, shared ride services and low-speed automated shuttles are just two options already available to varying degrees, aiding adults in their commuting needs [56]. Ultimately, while the number of older adults needing to engage in work may greatly increase, it will be some time before readily available alternatives to private transportation in many countries can provide adults with transportation needs so they can access the vast majority of jobs.

Second, for those who must continue to drive to work and for whom public transportation is not an option, the advances in automation are projected to increase the number of years that the individual can remain at work and on the same job [57]. For example, intersection crashes are a major risk for older drivers. Vehicles equipped with intersection assist or more advanced vehicle–to–vehicle communications can help greatly reduce such crashes for all older drivers. For drivers with neurodegenerative impairments (e.g., Parkinson’s disease), vehicles that can manage the lateral and longitudinal control of the vehicle, actively or passively, can be of real benefit. For drivers with arthritis of the neck and shoulders, center-mounted displays of side-view mirrors and back-up cameras can make it possible for them to continue driving. For drivers with vision problems, new smart headlamps can make night driving much safer.

Third, and perhaps most critically, advances in automation can potentially extend the number of years that older adults can continue driving commercial vehicles, something especially important given the shortage of drivers in some countries [58]. For instance, fatigue is a major contributor to crashes, especially those involving older drivers. With the advent of automated technologies, fatigue risk management systems can be deployed in real time [59]. New ways of monitoring drowsiness in the vehicle are also possible and can potentially be tailored to the needs of the driver. For the future, automation under development holds the promise of being able to predict microsleeps. When considered in combination, automation technologies can keep drivers and others on the road safe even during extraordinary events, such as when a driver has a medical event. The bottom line for older adults who drive to work, whose work is driving, or who take some form of public transportation is that they will be able to continue to drive safely for a much longer period of time.

Moreover, automation may be integral to overall work continuance. As stated previously, the ability to maintain an active working life depends on functional capacity. Providentially, advances in automation typically enhance functional capacity. Furthermore, advances in transportation promise to generate work environments that foster decent work, which is important for encouraging workers to remain in the workforce longer. Additionally, as has been shown in the recent COVID-19 pandemic, working from home may be considered a realistic continuing way of working.

## 3. Gaps in the Literature about Aging and Work

While there is a rich literature that describes aging and work ability, there are still areas that require consideration and investigation. We identified and describe five gaps next.

### 3.1. Early Work Disability Studies

In emerging and developing countries, particularly in Latin America, a number of cross-sectional studies investigated work ability and impacts on early functional aging [60,61]. However, further population-based studies with a longitudinal design are required [3] in order to address potentially related risk factors and to examine for differences based on sociocultural characteristics that may be associated with work ability. This knowledge could support specific actions to protect work ability.

### 3.2. Population Dashboards of Workers’ Health Metrics

Population-level data dashboards of workers’ health metrics have recently been made available, such as the Global Burden of Disease Study [62], the SmartLab Brazilian Observatory of Occupational Health and Safety [63], and the European Working Conditions Survey [64]. Notwithstanding, these resources are scarce and/or insufficient to support a rigorous assessment of the strategic factors pertinent to workers’ health. In addition, there are limitations in official records about factors that may negatively impact work ability. Tools that consolidate measures of work ability, occupational features, and extra-occupational features are suggested to support situational diagnostics by directing preventive actions.

### 3.3. Intervention and Cost-Effectiveness Studies in Health Promotion and Prevention of Early Functional Aging at Work

Efforts should be made to identify highly effective low-cost programs that represent the gold standard or paradigm of successful actions to improve work ability, thereby reducing the likelihood of premature functional aging. Such programs should not only seek returns on investment but also add value as an end goal (value on investment).

Few workplace health promotion programs are the subject of a careful process of evaluating their financial impact and individual and organizational benefits [65]. A recent meta-analysis conducted by Oakman and colleagues [22] showed a positive effect of the workplace interventions improving work ability, but the evidence is moderate according to the Grading of Recommendations Assessments, Development of Evaluation criteria, and the results are inconclusive, indicating the need for further studies [66].

### 3.4. Policies for Tailoring Work Demands to Individual Needs and Abilities

In order to maintain work ability, it is necessary to address individual circumstances. Therefore, it is recommended to implement policies aimed at assessing and adjusting the balance between work demands and individual resources that aid in meeting the demands, due to aging-related functional difficulties, including burnout, as well as to consider the specificities of each age group [3,21,22,24].

Ilmarinen and colleagues [50] highlighted the knowing–doing gap, which describes the divergence between knowledge gains and ability to carry out successful actions, as well as the length of time it takes to perform the actions. One way of giving visibility to the promotion of workers’ health and safety and the prevention of functional aging at work is to incorporate work ability measures into the institutional Balanced Scorecard approach to strategic occupational safety and health management [67].

### 3.5. Iniquities of Social Protections for Aging Workers

As newer state-mandated reforms of social security systems are implemented, such as the raising of the minimum retirement age enacted in numerous countries, workers will have to remain in the workforce longer. Given the constraints associated with aging (particularly those of physical and functional order), it is necessary to implement social protection mechanisms to enable the exit (voluntary or involuntary) of older workers from work [3,21,22].

When extending adults’ working lives is being considered, barriers and opportunities (e.g., legal and social) should be explored. Policies to maintain the employability of older people should work on many fronts, including skills upgrading, as well as improvements in work conditions and organization. In environments and sectors where there are difficulties, such as shortages in employment opportunities, legal and social actions that minimize limitations and possible discriminatory aspects against older workers should be prioritized [21,24,68].

## 4. Decent Work and Aging

As the length of working life increases, so does the challenge for decent work. The impact of longer working life on workers can be significant in both positive and negative ways. From a positive perspective, “work is the main source of income for consumption and savings. It serves as an anchoring function in society and can be a source of dignity and purpose” [69]. From a negative perspective, prolonged work years can increase morbidity and mortality from work-related injuries and illnesses and can cause longer recovery times, burnout, job lock, age discrimination, periods of unwanted employment, and less free time [70].

The promotion of decent work can be addressed by (1) taking a life span approach to aging; (2) using comprehensive efforts to address health, work ability, and job retention at an older age; (4) emphasizing that decent work requires attention to both the workers and the enterprise; and (5) providing a supportive culture for multiple generations. These elements have been described as components of a “productive aging” and “successful aging” approach, which can be instrumental in bringing about decent work [71,72]. This article has touched upon many of the elements of productive aging, but a few merit further exposition. Chief among these is that the optimum time to start to address the needs of older workers is in the decades that proceed later life. Promoting education and health throughout life is the best way of ensuring that workers can function and be employed when they are older [16,68,73].

From a lifespan perspective, it is necessary to realize that, throughout life, various mental and physical functions decline, but the nature of the decline differs among people [74]. A life course approach to prevention and management of occupational injuries and illnesses, particularly involving musculoskeletal problems, comorbidities, and multi-morbidities, can be effective in addressing the needs of older workers [75]. Moreover, as workers age, the burden of chronic diseases increases. Making work amenable to workers with controlled chronic disease is a necessary part of decent work [76,77].

Overall, there is a need for employers, organizations, and decision makers to create work environments that foster decent work for aging workers. In addition to how work is organized, this involves addressing the needs for maintaining and upgrading the skills of all workers, particularly mature-aged workers. While older workers can proficiently use new technologies and learn new skills, as the rate of technology diffusion increases, it may be more difficult for older workers to be prepared for jobs and job changes [78]. The higher-order cognitive skills, such as complex problem-solving, critical thinking, and advanced communication, which are transferable across jobs, build on experience and cannot be acquired by school alone [78].

Additionally, aging of the workforce overlaps with developing the “built environment”, which is critical for individuals functioning in workplaces, homes, and cities [79,80,81]. Places and spaces that promote worker activity and functioning can be instrumental in prolonging work life [82,83].

Finally, there are many factors that influence the intersection of aging and decent work. It is important that interventions focus not just on workers but also on the enterprises that employ them, because there is a reciprocal relationship between them. Workers benefit from organizational changes that support their aging, and organizations benefit from productive aging workers [23,81,84,85,86,87].

## 5. Conclusions

This paper examined how aspects of aging and organizational factors affect work ability across the lifespan and throughout one’s work career. Additionally, the critical issue of worker physical mobility was also addressed as a practical limitation to functional aging. Through our investigation we identified gaps in the literature where research and interventions should be promoted. In conclusion, there is a need for employers, organizations, and decision makers to create work environments that foster decent work for aging workers.

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
