# Peer review of "Aging and the Future of Decent Work"

_ijerph, 2021, doi:10.3390/ijerph18178898_

Round 1
Reviewer 1 Report
Ageing and The Future of Decent Work:
This manuscript is a timely and suggestive paper on job creation for the elderly. This paper examined how ageing and organizational factors affect workability across the lifespan. This identified gaps in the literature where research and interventions.
However, (1) it is necessary to quantify the data with lots of literature and (2) have statistical validation data. Also, (3) please provide empirical examples.
Reviewer 2 Report
This paper is written in the format of a commentary piece refleting authors' experience and perspectives on the aging and the future of decent work. In general the commentary is written well and authors' opinions supported by relevant literature. The commentary piece can be accepted for publicationas it is. However, I also suggest a careful language editing to improve the flow of the ideas and removal of grammatical minor issues and typos.
Reviewer 3 Report
First of all, congratulations to the authors for their work. It is an interesting manuscript that reflects and reflects on decent work and ageing, at a crucial time for such an important issue in the present and in the future. A few comments are noted that could add some ideas to the text.
KEYWORDS
It would be interesting to add the words “aging” and “decent work”
INTRODUCTION
Pag 1 line 32: I believe that the expression "declining fertility" places a greater emphasis on biological causes than on the social causes of the reality of declining birth rates, which can be misleading.
Pag 2 line 49-53: The UN definition of decent work is adequate, but perhaps adding another definition from the scientific literature may help to give a more complete theoretical framework to the concept and help the reader as well.
- WORK ABILITY AND ….
Pag 2 line 68-70: Check, there are spaces between words that have been added by mistake.
Pag 2 line 74-78: It would be interesting to be able to relate which of these effects is greater in individuals, institutions or society, and to relate this in turn to the different quotes that are pointed out.
Pag 2 line 84: It’s “alone” the best option in this phrase?
2.1. ORGANIZATIONAL FACTORS AND WORK ABILITY
Pag 2 line 88-92: The way the first sentence is expressed contradicts the way this paragraph closes. It would be interesting to rework the writing, to make it clear that it is the better or worst working conditions that make works an element that is good or not good for health (if this is what the authors really want to express with this paragraph).
2.2. AGING AND THE CHANGING….
Pag 4 line 157: Remove the first "for"
- GAPS IN THE LITERATURE
Pag 5 line 232: The numbering of this heading is incorrect, it would be 3.3 (in the next heading too)
In general: Could you add another aspect such as reduced working hours for older workers? Or gradual retirement as an issue to be explored especially in ageing societies?
Elsayed, A., de Grip, A., Fouarge, D., & Montizaan, R. (2018). Gradual retirement, financial incentives, and labour supply of older workers: Evidence from a stated preference analysis. Journal of Economic Behavior & Organization, 150, 277-294
Round 2
Reviewer 1 Report
Moderate English changes required.